# Long Non-Coding RNA LINC02802 Regulates In Vitro Sprouting Angiogenesis by Sponging microRNA-486-5p

**DOI:** 10.3390/ijms23031653

**Published:** 2022-01-31

**Authors:** Stefania Rosano, Sushant Parab, Alessio Noghero, Davide Corà, Federico Bussolino

**Affiliations:** 1Department of Oncology, University of Torino, 10124 Orbassano, Italy; sushant.parab@unito.it (S.P.); federico.bussolino@unito.it (F.B.); 2Candiolo Cancer Institute, IRCCS-FPO, 10060 Candiolo, Italy; 3Lovelace Biomedical Research Institute, Albuquerque, NM 87108, USA; anoghero@lovelacebiomedical.org; 4Department of Translational Medicine, Piemonte Orientale University, 28100 Novara, Italy; davide.cora@uniupo.it; 5Center for Translational Research on Autoimmune and Allergic Diseases—CAAD, 28100 Novara, Italy

**Keywords:** sprouting angiogenesis, lncRNAs, microRNAs, ceRNA

## Abstract

In the last several years, accumulating evidence indicates that noncoding RNAs, especially long-noncoding RNAs (lncRNAs) and microRNAs, play essential roles in regulating angiogenesis. However, the contribution of lncRNA-mediated competing-endogenous RNA (ceRNA) activity in the control of capillary sprouting from the pre-existing ones has not been described so far. Here, by exploiting the transcriptomic profile of VEGF-A-activated endothelial cells in a consolidate three-dimensional culture system, we identified a list of lncRNAs whose expression was modified during the sprouting process. By crossing the lncRNAs with a higher expression level and the highest fold change value between unstimulated and VEGF-A-stimulated endothelial cells, we identified the unknown LINC02802 as the best candidate to take part in sprouting regulation. LINC02802 was upregulated after VEGF-A stimulation and its knockdown resulted in a significant reduction in sprouting activity. Mechanistically, we demonstrated that LINC02802 acts as a ceRNA in the post-transcriptional regulation of Mastermind-like-3 (MAML3) gene expression through a competitive binding with miR-486-5p. Taken together, these results suggest that LINC02802 plays a critical role in preventing the miR-486-5p anti-angiogenic effect and that this inhibitory effect results from the reduction in MAML3 expression.

## 1. Introduction

Sprouting angiogenesis (SA) is the biological process that leads to the formation of new blood capillaries from pre-existing ones. SA is a complex multistep event and requires in its different phases an orchestrated control of many cellular functions [1]. Such regulation is achieved by the combined action of many regulatory molecules together with the post-transcriptional modulation of gene expression provided by microRNA (miRNAs) and long non-coding RNAs (lncRNAs) [2]. Among numerous endogenous factors with pro- or anti-angiogenic effects [3], vascular endothelial growth factor A (VEGF-A) is considered to be the master regulator of SA, acting both as a strong migratory and mitogenic signal [4]. Under physiological conditions, a perfect balance of pro- and anti-angiogenic inducers is maintained and ECs remain in a state of quiescence. Upon the release of pro-angiogenic factors, ECs undergo activation to participate in the ordered formation of the vascular net in crucial processes, such as embryonic development, organogenesis, female reproductive cycle, lactation and muscle hypertrophy [1,5]. Uncontrolled angiogenesis is instrumental to various disorders, including cancer progression and metastasis spreading; in these pathological conditions, the loss of the homeostatic balance between the pro- and anti-angiogenic signals results in improper angiogenesis characterized by the formation of abnormal, poorly functional capillaries [1,5].

The importance of miRNAs in endothelial physiology and angiogenesis has been demonstrated in vitro and in vivo by the EC-specific inactivation of the enzymes Dicer and Drosha, which are responsible for the maturation of miRNAs [6,7], and, recently, it has been demonstrated that the complete depletion of miRNAs suppresses tumor angiogenesis [8,9]. Currently, around 33 miRNAs belonging to different miRNA families have been described as angiogenesis regulators. In a previous work, we demonstrated that, in the initial step of SA, miRNAs act cooperatively to support the specification of the tip cell phenotype by reducing the expression of genes that are associated with VEGF-A-mediated cell proliferation while de-repressing genes that are involved in cell migration and extracellular matrix remodeling [10].

There is increasing evidence demonstrating that other non-coding RNAs, such as lncRNAs, also play a significant role in the regulation of angiogenesis. These RNAs are usually longer than 200 nucleotides, exhibit a distinct biogenesis from that of protein-coding RNAs, are poorly conserved along species and show dynamic and specific patterns of expression in tissues and cells [11,12]. Although the biological relevance of lncRNAs remains largely unknown, they have emerged as essential mediators of several cellular processes, and the deregulation of their expression has been associated with various diseases, including cancer [13]. LncRNAs are highly versatile [14]; depending on their localization and interactions with other RNA and proteins, lncRNAs can modulate cellular activities differently. At the nuclear level, lncRNAs influence the chromatin assembly and can regulate the activity of a chromosomal locus or that of a specific gene. Cytosolic lncRNAs modulate RNA stability and protein translation and act as competing endogenous RNAs (ceRNAs) by binding specific miRNAs, thus altering their availability for other cellular targets. The final result of this regulation is an increase in the expression of the miRNA target genes [15,16,17]. This miRNA-sponge activity was shown to be important in many cellular processes during development [18,19] and in several diseases, including cancer [20,21]. Although the importance of lncRNAs in the vascular unit has been known for several years, a limited number of lncRNAs have been described to play a role in both physiologic and pathological conditions [2,22,23], and some cases of the miRNA-sponge effect have also been reported [24,25,26,27,28]. However, the contribution of lncRNA-mediated ceRNA activity in the context of the regulation of the SA process has not been described so far.

By exploiting gain-of-function and loss-of-function approaches, in this study, we identified LINC02802, an intergenic lncRNA that is so far not associated with any biological function, as an in vitro regulator of SA. LINC02802 was upregulated in an in vitro model of SA after VEGF-A stimulation. Mechanistically, we demonstrated that LINC02802 acts as a ceRNA in the post-transcriptional regulation of Mastermind-like 3 (MAML3) gene expression through the competitive binding of miR-486-5p. Furthermore, our study reveals a potential new role for MAML3 in the sprouting process.

## 2. Results

### 2.1. Identification of lncRNAs Involved in the Regulation of SA

The expression and function of a limited numbers of long non-coding RNAs (lncRNAs) has already been described in vasculature [29,30]; however, a comprehensive analysis of lncRNAs involved in the regulation of the process of sprouting angiogenesis (SA) has not yet been specifically reported. By exploiting a consolidate three-dimensional model that mimics SA in vitro, we previously generated a complete transcriptomic profile of ECs during the sprouting process triggered by VEGFA [10]. Spheroids challenged with VEGF-A for 18 h (SPHV) and control spheroids (SPHC) were processed for long- and small-RNAseq. In this RNA-Seq dataset, seventy-five lncRNAs with a TPM > 1 that were differentially expressed between SPHC and SPHV were identified (Figure 1A); fifty-six of them were upregulated and nineteen downregulated. The complete list of name of DE lincRNAs is reported in the Appendix A. An interesting function of lncRNAs, still poorly characterized in angiogenesis, is the modulation of the miRNAs availability through the ceRNA effect. Therefore, we decided to look for, among our differential expressed lncRNAs, some possible lncRNAs involved in sponge circuits. To investigate miRNA–lncRNA–gene circuits, we primarily focused on the set of differentially expressed lncRNAs between SPHV and SPHC spheroids, and we first ranked them according to the mean expression level within our experimental conditions. The top five lncRNAs with a higher base mean expression were MEG3, AC007998.3, LINC02802, AC006453.3 and PCAT19 (Figure 1A), from which, LINC02802 had the highest log2 fold change value. LINC02802 is a long intergenic ncRNA (lincRNA) transcribed from the reverse strand of chromosome 1. Its function is completely uncharacterized. As shown in Figure 1B, LINC02802 transcription was significantly increased in SPHV compared to SPHC, suggesting an important role for this lncRNA in the sprouting process.

We next sought to investigate the role of LINC02802 in the sprouting process by exploring its network of possible interactors in terms of its sponge activity on miRNAs in particular. For this task, we explored, through a bioinformatic approach, the possibility of differentially expressed miRNAs between SPHV and SPHC (Figure 1C) interacting with LINC02802 by a combination of sequence-based predicting algorithms (see Materials and Methods). The complete list of name of DE microRNAs is reported in the Appendix A. From our analysis, we found LINC02802 putatively targeted by four different miRNAs, out of which, two were differentially expressed: hsa-miR-486-5p (downregulated with a log2FC: −0.82) and hsa-miR-874-3p (upregulated with a log2FC: 0.67). In particular, hsa-miR-486-5p had a higher MiRanda binding score (158) and RNAhybrid_mfe (−38.4) compared to the latter one, suggesting the interaction between hsa-miR-486-5p and LINC02802 to be more stable. Since the expression of miR-486-5p was significantly downregulated in EC spheroids after VEGF-A stimulation (Figure 1D), LINC02802 and hsa-miR-486-5p turned out to be negatively correlated in terms of the expression between the SPHV and SPHC conditions, and in agreement with the usually repressive activity exhibited by miRNAs on their target mRNAs.

Finally, to completely define a single putative sponge circuit driven by LINC02802, we looked for the protein coding genes targeted by hsa-miR-486-5p. According to Targetscan analysis, 107 different protein coding genes expressed in ECs were potentially targeted by hsa-miR-486-5p; a complete list of them is provided in Appendix A. Targetscan predictions indicated that hsa-miR-486-5p targeted MAML3 with a high weighted context++ score percentile. Through literature inspection, among the possible targets, MAML3 was also the most relevant target gene in our biological context, according to its reported involvement in the NOTCH pathway [31]. Therefore, the circuit selected included LINC02802–miR-486-5p–MAML3..

### 2.2. Silencing of LINC02802 Represses Sprouting in EC Spheroids

To investigate whether LINC02802 might play a role in SA, its expression was altered in ECs by overexpression and silencing approaches. RT-qPCR was exploited to confirm the over-expression efficiency of the vector pcDNA3.1-LINC02802 (Appendix A) and to select siRNA siLINC02802-2 as the most effective silencing sequence (Appendix A). As shown in Figure 2A, the over-expression of LINC02802 did not affect sprouting. On the contrary, the silencing of LINC02802 resulted in a significant reduction in sprouting activity (Figure 2B). These data indicate that LINC02802 expression is necessary to support the ECs sprouting process. ceRNA activity is a prerogative of cytoplasmic lncRNAs and the Encode public database classifies LINC02802 as a cytoplasmic lincRNA. In order to validate this indication in ECs, we performed a separation of the nuclear and cytosolic compartment; we performed a RT-qPCR assay on isolated poly-adenylated RNA, confirming that LINC02802 predominantly localized in the cytosolic compartment. As a control for this technique, we evaluated the subcellular localization of MEG3, which is known to be nuclear [32] (Figure 2C).

### 2.3. LINC02802 Acts as a Competitive Endogenous RNA for miR-486-5p

Next, we investigated whether the mechanism of action of LINC02802 could rely on ceRNA activity for miR-486-5p, as suggested by the in silico analysis. The direct interaction between these RNAs was analyzed by an evaluation of the impact of miR-486-5p modulation on endogenous LINC02802 expression levels. We found that miR-486-5p over-expression significantly downregulated LINC02802 (Figure 2D), whereas the knockdown of miR-486-5p significantly increased LINC02802 expression (Figure 2E). The effectiveness of the over-expression and knockdown of miR-486-5p was evaluated by RT-qPCR (Appendix A), which demonstrated that miR-486-5p was upregulated 40,000 times or downregulated 10 times, respectively. To confirm our prediction, a dual luciferase assay was performed to validate the direct interaction between LINC02802 and miR-486-5p. The wild type LINC02802 sequence and its mutated form, which was defective of the miR-486-5p binding site (Figure 2F), were cloned downstream the luciferase gene into a pMIRGLO dual-luciferase vector. As shown in Figure 3G, the co-transfection of the miR-486-5p mimic with pMIR-LINC02802 WT leads to a significant reduction in luciferase activity. On the contrary, in cells co-transfected with the mutated form of LINC02802 (pMIR-LINC02802 MUT), the inhibitory effect on luciferase activity was lost. Collectively, these results confirmed that miR-486-5p negatively regulated LINC02802 expression through direct interaction.

### 2.4. LINC02802 Supports SA by Counteracting the Anti-Angiogenic Effect of miR-486-5p

ECs were then transfected with a miR-486-5p mimic/inhibitor or with the corresponding controls and used to perform the sprouting assay. As expected, miR-486-5p over-expression significantly inhibited the sprouting of ECs (Figure 3A), whereas miR-486-5p inhibition did not affect the process (Figure 3B). To obtain further evidence of the interplay between miR-486-5p and LINC02802 on SA, we analyzed the expression of miR-486-5p after the knockdown and over-expression of LINC02802. As shown in Figure 3C, and in agreement with the ceRNA hypothesis, LINC02802 silencing increased the expression of miR-486-5p. Conversely, LINC02802 over-expression reduced miR-486-5p levels, but only when LINC02802 contained the intact binding site of miR-486-5p (Figure 3D). The evidence that LINC02802 negatively regulated miR-486-5p expression raised the question of if LINC02802, by controlling miR-486-5p levels in spheroids during the sprouting process, could prevent its anti-angiogenic effect. To test this possibility, we investigated the effect of LINC02802 over-expression on the anti-angiogenic effect mediated by miR-486-5p mimics. As shown earlier, the over-expression of miR-486-5p inhibited sprouting. However, the co-transfection of LINC02802 considerably restored sprouting activity (Figure 3E).

These data suggest that miR-486-5p has an anti-angiogenic effect on spheroids, and that LINC02802 and miR-486-5p exert an opposite effect in the regulation of the sprouting process. Therefore, taken together, these results support our hypothesis that LINC02802, acting as a sponge for miR-486-5p, plays a critical role in sprouting preventing the miR-486-5p anti-angiogenic effect and favoring the SA process.

### 2.5. miR-486-5p Directly Targets MAML3, Modulating Its Expression

Our bioinformatic analysis based on Targetscan predictions indicated MAML3 as the protein coding gene involved in the LINC02802-miR-486-5p sponge circuit. To confirm this information, we first analyzed MAML3 expression in the SA model. As shown in Figure 4A,B, we found an increased expression of MAML3 after VEGF-A stimulation, which might indicate a negative correlation between miR-486-5p and MAML3 expression. Sequence analysis showed one conserved 8mer miR-486-5p binding element and one poorly conserved 7mer site for miR-486-5p on MAML3 3′-UTR (Figure 4C). In order to test whether miR-486-5p could post-transcriptionally regulate MAML3 by direct binding within the 3′-UTR region, a dual luciferase reporter assay was performed in ECs. As shown in Figure 4D, the co-transfection of the miR-486-5p mimic with pMIR-MAML3 WT significantly repressed luciferase activity. This inhibitory effect failed when ECs were transfected with the MAML3 mutant (pMIR-MAML3 MUT). We next sought to determine whether miR-486-5p can repress endogenous MAML3. To this end, we modulated the expression of miR-486-5p with the specific mimic and inhibitor and then examined MAML3 expression by RT-qPCR and Western blot analysis. As expected, the miR-486-5p mimic significantly repressed MAML3 expression (Figure 4E), whereas the miR-486-5p inhibitor markedly increased MAML3 at mRNA and protein levels (Figure 4). These data indicate that miR-486-5p directly targets MAML3 and represses its expression.

### 2.6. MAML3 Over-Expression Counteracts the Inhibitory Effect of miR-486-5p on SA

To clarify the functional role of MAML3 in the sprouting process, loss- and gain-of-function approaches were used. The efficiency of these manipulations was monitored by RT-qPCR and a Western blot (Appendix A). We found that MAML3 silencing resulted in a strong reduction in endothelial sprouting (Figure 5A), in agreement with the phenotype obtained with the over-expression of miR-486-5p. On the contrary, MAML3 over-expression did not modify sprouting activity (Figure 5B). Furthermore, the co-transfection of MAML3 with the miR-486-5p mimic counteracted the inhibitory effect of miR-486-5p overexpression, thus rescuing the sprouting process (Figure 5C). In summary, these data indicate that MAML3 is essential to SA and that the inhibitory effect on sprouting observed after miR-486-5p over-expression could be due to a reduction in MAML3 expression.

### 2.7. LINC02802 Indirectly Modulates MAML3 Expression

Finally, to confirm that LINC02802 act as a ceRNA and to evaluate the effect of the interaction between LINC02802 and miR-486-5p on MAML3, we examined the expression of MAML3 both at transcript and protein level following the manipulation of LINC02802 expression. We found that the over-expression of LINC02802 resulted in an increased expression of MAML3, but only when the miR-486-5p binding site was intact (Figure 6A). On the contrary, the knockdown of LINC02802 resulted in a decreased expression of MAML3 (Figure 6B). Overall, these results indicate that lincRNA LINC02802, acting as a ceRNA for miR-486-5p, affects the abundance of miR-486-5p and indirectly impacts MAML3 expression.

## 3. Discussion

SA is a dynamic process in which ECs migrate, form lumens and organize a capillary network that is remodeled into a hierarchically branched and functionally perfused vascular bed. In the nascent sprout, ECs adopt two distinct cellular phenotypes, known as tip and stalk cells, with specialized functions and specific gene expression patterns [33]. VEGF-A and the NOTCH ligands DLL4 and Jagged-1 are the external cues that regulate the dynamics of the tip–stalk phenotype. Analysis of the transcriptional profile of retinal ECs isolated from Dll4 heterozygous null mice, which present an excessive numbers of tip cells, indicates an enrichment in the tip cells of genes that regulate the interaction with the extracellular matrix and code for secreted molecules, some of which regulate stalk cell behavior [34]. The specific transcriptional profile of tip cells was recently confirmed by single cell RNA-Seq performed in a laser-induced choroid neovascularization model and in endothelial cells isolated from lung cancer [35,36]. A precise gene expression pattern in stalk cells has yet to be clearly defined [35], but it is likely to be regulated by transcriptional programs triggered by Hey/Hes transcription factors, which are down-stream to NOTCH activation [37]. At systems level, there is little information on the features of the transcriptional landscape characterizing SA. A specific mRNA network obtained by siRNA screening has been recently described in bi-dimensional migrating vascular and lymphatic ECs [38]. During the migratory phase of in vitro SA, the global analysis of miRNA–mRNA interactions identified two sub-network modules, the first organized in upregulated miRNAs connected with downregulated target genes and the second with opposite features [10].

Here, by exploiting an in vitro three-dimensional SA assay, which mainly allows us to analyze the tip migratory phenotype, we characterized the expression pattern of lncRNAs in human ECs undergoing sprouting, which represents a step ahead of previous global lncRNAs annotations of endothelial analyzed in two-dimensional conditions [29,30]. We found 75 differentially expressed lncRNAs in the spheroid model. Interestingly, the large majority of these lncRNAs were not described before as being related to angiogenesis. Therefore, our data could represent an important starting point to dissect the relevance and functions of this class of regulatory molecules in the sprouting process. This observation is well supported by the large amount of evidence that lncRNA expression patterns are preferentially context-specific [39,40], and that the anatomical origin and the functional state of ECs dictates the transcriptional landscape, as demonstrated by single cell analyses [35,36,41].

Having previously demonstrated through a global miRNA–mRNA analysis that miRNA activity facilitates the phenotypic transitions described in SA [10], and since lncRNAs can compete for miRNA binding [15,16,17], we analyzed the extent of the sponge effect and ceRNA circuits exerted by lncRNAs that were differentially expressed in VEGF-A-stimulated spheroids compared to resting EC spheroids. Among all the differentially expressed lncRNAs, the best candidate in terms of the expression rate, log2 fold change and novelty was LINC02802. We demonstrated that LINC02802 was upregulated during SA and that its expression is required to support this process, as inferred by the inhibitory effect exerted by LINC02802 silencing. Mechanistically, we have shown that lncRNA LINC02802 exerted a sponge effect on miR-486-5p and that LINC02802 expression prevented MAML3 degradation by sequestering miR-486-5p (Figure 7), ultimately favoring the sprouting process. We also found that the knockdown of LINC02802 significantly increased the expression of miR-486-5p, whereas LINC02802 over-expression reduced its level. On the other hand, the miR-486-5p mimic reduced LINC02802 expression and the miR-486-5p inhibitor increased the LINC02802 level, providing supporting evidence that LINC02802 and miR-486-5p reciprocally modulate their expression. Accordingly, miR-486-5p was down-modulated in sprouted ECs, demonstrating a negative correlation between LINC02802 and miR-486-5p in this model of SA, thus supporting the sponge hypothesis. This effect favors SA, as deduced by the observation that miR-486-5p over-expression significantly impaired sprouting, which was blunted by increasing LINC02802 levels. Moreover, the inhibition of sprouting induced by LINC02802 knockdown was partially reversed by miR-486-5 down-modulation.

miR-486-5p is known to be a versatile microRNA and has been reported to have central roles in several types of cancers and other non-oncological conditions, such as autism, intervertebral disc degeneration and metabolic syndrome [42]. In cancer, miR-486-5p was found to have a bivalent role, acting as a tumor suppressor in some cancer types, such as breast [43], or showing tumorigenic activity in prostatic cancer [44].

The data here reported uncover, for the first time, a possible function of miR-486-5p in ECs, and describe its cooperation with LINC02802 in a regulatory loop. Targetscan predictions, biologically validated in our model, identified the MAML3 transcript as the target gene of this ceRNA mechanism. In the SA model, the amount of MAML3 negatively and positively correlated with miR-486-5p and LINC02802 expression levels, respectively. Accordingly, the over-expression of miR-486-5p suppressed MAML3 expression, whereas the downregulation of miR-486-5p enhanced the MAML3 level. At the functional level, we found that the silencing of MAML3 resulted in a strong reduction in sprouting, in accordance with the phenotype obtained by over-expressing miR-486-5p and silencing LINC02802. Consequently, MAML3 over-expression restored the inhibitory effect of miR-486-5p on SA. Mirrored data confirmed the regulatory role of LINC02802, that, when over-expressed or silenced, increased and reduced the expression of MAML3, respectively. Collectively, the over-expression of miR-486-5p or the silencing of LINC02802 resulted in a deficiency of MAML3 expression and, as a consequence, in the inhibition of sprouting.

Members of the MAML gene family (MAML1-3) are characterized by an N-terminal basic domain that binds to the ankyrin repeat domain of Notch, thus contributing to its biological activities [31]. Increasing evidence indicates that MALMs are pleiotropic interactors across multiple signaling pathways, including Wnt/β-catenin, Sonic Hedgehog and Hippo [45]. Of note, MALM3 binds to Notch less efficiently and shows a weaker transactivating activity on the promoter of the Notch target HES1 compared to MALM1 and MALM2 [46]. Furthermore, the SA model utilized generates mostly tip cells, which are known to activate the Notch pathway in adjacent stalk cells by expressing DLL4 on their membrane [47,48,49]. In fact, in our previous work we demonstrated that the spheroid model was unable to properly differentiate stalk cells. We also observed an increase in SA in the presence of the γsecretase inhibitor DAPT, which inhibits Notch endoproteolysis and activation [10]. These observations suggest that MAML3 is most likely not involved in Notch activation in our model and raise the question of what the function of MAML3 could be in the genetic program that sustains the tip phenotype. The analysis of miRNAs and miRNA target genes in the peripheral blood mononuclear cell of baboons fed with an atherogenic diet showed that VEGFA, MAML3 and LRP2 are the major hubs that are differentially expressed between low and high LDL-C baboons and inversely expressed with respect to miRNAs that are differentially expressed in low and high LDL–C baboons [50]. From this observation, it could be interpreted that this network should be important in the cross talk between inflammatory monocytes and ECs in the onset of artherogenesis in determining the plaque instability, which is characterized by a prominent angiogenesis [51]. In neuroblastoma, it has been reported that MAML3 over-expression increased IGF2 transcription independently from Notch, and this mediated IGF1R activation and AKT signaling, resulting in an increased cell proliferation [52]. Interestingly, as reported, previous works also demonstrated that miR-486-5p is involved in the regulation of the IGF2/IGF1R pathway by targeting IGF1R [53]. The importance of the IGF/IGFR pathway in angiogenesis has already been demonstrated. IGF/IGFR predominantly act as pro-angiogenic signaling through the stimulation of VEGF-A, hypoxia-inducible factor 1α (HIF1α) and endothelial nitric oxide synthase expression [54]. In ECs, IGF2 promoted migration, tube formation and ECs invasion through an increased MMP2 expression, but had no effect on proliferation; additionally, IGF2 increased p38 MAPK expression rather than having an effect towards cell proliferation [55]. Based on this data, we could speculate that LINC02802, acting as a molecular sponge for miR-486-5p, may be involved in the regulation of SA sustained by the IGF2/IGF1R pathway.

## 4. Materials and Methods

### 4.1. Primary Cell Culture

Human umbilical vein endothelial cells (HUVECs) were isolated by collagenase digestion of the interior of the umbilical vein of cords from different donors, as previously described [56], and used between passage 1 and 3. To minimize experimental variability due to individuals’ different genetic background, each HUVEC batch was composed by cells derived from 3 to 5 different cords, irrespective of their sex. HUVECs were cultured in M199 medium (Sigma, St. Louis, MA, USA) supplemented with 20% fetal bovine serum (FBS) (Sigma, St. Louis, MA, USA), 0.2% brain extract, 0.05 μg/mL porcine heparin (Sigma, St. Louis, MA, USA) and 2% penicillin–streptomycin solution (Sigma, St. Louis, MA, USA). Collection of umbilical cords is governed by an agreement between Università degli Studi di Torino and the “Ordine Mauriziano di Torino” hospital, protocol number 5 2 September 2014. Informed consent was obtained from all subjects involved.

### 4.2. Spheroid Capillary Sprouting Assay

EC spheroids were generated as described [10,56], with minor modifications. HUVECs within the third passage were trypsinized and cultured in hanging drops (800 cells/drop) in M199 containing 10% FBS and 0.4% (*w*/*v*) methylcellulose. After overnight incubation, spheroids were collected and embedded in a solution containing 15% FBS, 0.5% (*w*/*v*) methylcellulose, 1 mg/mL collagen solution, 30 mM HEPES and M199 from 10× concentrate. A total of 0.1 M NaOH was added to adjust the pH to 7.4 and to induce collagen polymerization. Sprouting was induced by addition of 20 ng/mL recombinant human VEGF-A (R&D Systems, Minneapolis, MN, USA) to the collagen solution. Spheroids were imaged or collected for further analysis after 18 h incubation at 37 °C in a 5% CO_2_ incubator.

### 4.3. Imaging

Spheroids in the 3D collagen matrix were fixed with 4% PFA for 30′ at room temperature and imaged using an AF6000LX-TIRF Workstation (Leica, Wetzlar, Germany). Measurement of the total area of the sprouts per spheroid was performed on phase contrast images using the Image J software package. For each experimental condition, the sprout area of at least 20 spheroids was measured.

### 4.4. RNA Isolation

Total RNA was isolated from cells or spheroids with Maxwell RSC miRNA Tissue kit (Promega, Madison, WI, USA). Quality and concentration of RNAs were assessed with a NanoDrop ND-1000 spectrophotometer (ThermoFisher Scientific, Waltham, MA, USA).

#### RNA-Sequencing Analysis

The total RNA and small RNA analysis were performed as previously described [10]. Raw reads from the total RNA sequencing (GSE115817) were mapped to the human reference genome (hg19) by STAR-2.5 aligner [57]. Raw reads from the small RNA sequencing (GSE115954) were aligned to mature miRNAs sequences (miRbase v.22) by the BWA tool [58]. Annotations provided by the Ensembl database were set as reference for the RSEM [59] computational pipeline used for quantification of gene expression levels, while the SAMtools suite was used for miRNAs quantification [60]. Here, we specifically concentrated on annotated “lincRNA” Ensembl genes (7845 in total), from which, 168 were expressed with TPM > 1 in our RNA sequencing data and were thus retained for further analysis. Differentially expressed genes (DEGs) were calculated using DESeq2 [61]. Genes or miRNAs that met the criteria of |log2FC| > 0.5 and FDR < 0.05 were defined as differentially expressed.

### 4.5. Target Prediction

Regulatory relationships between miRNAs and target mRNAs were investigated separately for protein-coding genes and lncRNAs. Since lncRNAs are usually poorly conserved across species [62], target prediction based on conservation is less reliable; thus, for miRNA–lncRNAs target prediction, we employed a combined use of predictive algorithms not based on sequence conservation [63,64]. Taking these considerations into account, we used MiRanda-3.3a [65] and RNAhybrid-2.1.2 [66] for predicting miRNA targets on lncRNAs. MiRanda uses a weighted dynamic programming algorithm (Smith–Watermann) for predicting targets between a set of mature miRNAs and a target mRNA. The target prediction method for RNAhybrid is instead based on calculations of mRNA secondary structure and energetically favorable hybridization between miRNA and target mRNA. The set of predicted miRNA–lncRNA targets were filtered for a MiRanda score > 150 and RNAhybrid mfe < −30 to define the final set of miRNA–lncRNAs pairs. Unlike lncRNAs, most of the miRNA–protein coding gene targets are conserved across species [67]. Therefore, we used TargetSCAN.v.7.1 [68], which identifies target sites with emphasis on sequence conservation and perfect base pairing on miRNA seed region. Protein-coding predicted targets of miRNAs with a weighted context++ score percentile > 50 were further used.

### 4.6. Quantitative Reverse Transcription PCR (RT-qPCR)

To analyze lncRNA or protein-coding gene expression levels, 1 μg of DNAse-treated RNA was reverse-transcribed by using High Capacity cDNA Reverse Transcription kit and random primers (ThermoFisher Scientific, Waltham, MA, USA). To analyze miRNA expression levels, 350–700 ng of DNAse-treated RNA was reverse-transcribed by using High Capacity cDNA Reverse Transcription kit in the presence of an miRNA-specific primer (ThermoFisher Scientific, Waltham, MA, USA). cDNA amplification was performed using gene- or miRNA-specific TaqMan assays and TaqMan PCR Universal MasterMix (ThermoFisher Scientific, Waltham, MA, USA) in a CFX96 thermocycler (Bio-Rad, Hercules, CA, USA). Each assay was run in triplicate. Relative gene expression was calculated by using the comparative CT (threshold cycle number) method [69] and expression of TBP or RNU44 was used as endogenous control. The following TaqMan Assays were used: TBP (Hs00427620_m1), MAML3 (Hs00298519_s1), LINC02802 (custom made from the thermoscientific tool), IGF2 (Hs01005964_g1), IGF1R (Hs00609566_m1), RNU44 (001094) and hsa-miR-486-5p (001278), all from ThermoFisher Scientific, Waltham, MA, USA.

### 4.7. Subcellular Localization of lncRNAs

To isolate total RNA and RNA from subcellular fractions, cells from a 15 cm cell culture dish were scraped with 1 mL of PBS. To obtain total RNA, samples were directly processed with Maxwell RSC miRNA Tissue kit (Promega, Madison, WI, USA). To obtain nuclear and cytosolic RNAs, cells were centrifuged at 4 °C for 3 min at 3000 rpm, washed once with PBS and then suspended in 400 µL of Buffer Dautry + EDTA (10 mM Tris-HCl pH 7.8, 140 mM NaCl, 1.5mM MgCl_2_, 10 mM EDTA, 0.5% NP40, 100 U/mL RNaseOUT) and incubated on ice for 5 min. Lysates were centrifuged at 4 °C for 5 min at 3000 rpm. Supernatant was immediately processed with Maxwell RSC miRNA Tissue kit (Promega, Madison, WI, USA) to obtain the cytosolic fraction. For the nuclear fraction, the pellet containing nuclei was washed once with Buffer Dautry + EDTA, centrifuged at 4 °C for 5 min at 3000 rpm, suspended in 200 µL of Buffer Dautry + EDTA and then processed with Maxwell RSC miRNA Tissue kit (Promega, Madison, WI, USA). The separation of nuclear and cytosolic fractions was performed by using Oligotex™ mRNA Spin-Column (QIAGEN, Germantown, MD, USA) according to the manufacturer’s instructions. Relative expression on cytoplasmic and nuclear fraction was expressed as percentage of total poly-adenylated RNA.

### 4.8. Plasmids 

The sequences encoding wild type and mutated LINC02802 (GCGCAGGCGC CCAGAGGCGC ACAGGAGACC TCAGGCCCAG ACTCCACTCC CCAGCTGTGA AAGGACTGCT GGCCAGACCC CCAAGCTAGC CCGCCAGGCC TCCATAGAGC TGCCCAGCAT GGCTGCATCC AGTACCAAGA GTTGGTGGGA GACGGGTGAG GTACAGGCTC AGTCTGCGGC CAAGACTCCG TCCTGCAAGA CTCTTTGGCA GTTGCACTGG GTACTGGCTG AGTTCTGGAA AACAGGAATG TGAGCAGAGT GCTGCAGATC TCCTTGTCAG GAGCACCTAA GAACTGGCAT GACTCTACAC CCTCTCTCCT TCTGCAATGA CCTCGTGTCA AGATGGCAGA CCCGCAGAGA TGAATGTGCC TGGATACCTG AGTCACCAGG TGGAGGAAAC TCCCATTGAC ATGCATCACA CTCCACATAT GAAACCCTTC TTTTTCTGCA CTGTCACTGG GGTTAGGGAT TATTACTCAC TGCAACTTAG CCTAGCATTA AGTTGCTTGA CTAATACAAG TTTCAATAAA TGTCGCTTCA TTTCCCATG) were synthesized and cloned into pcDNA3.1 plasmid by the GeneArt service (ThermoFisher Scientific, Waltham, MA, USA). The miR-486-5p binding site on LINC02802 was point-mutated as follows: GTACAGG to GTCCGGT. MAML3 full length sequence (NCBI Reference Sequence: NM_018717.5) in pReceiver-Lv105-Puro was purchased from GeneCopoeia and then cloned into pcDNA3.1 plasmid. MAML3 3′UTR was purchased from GeneCopoeia and mutagenized at the miR-486-5p recognition sites using the QuickChange Site-Directed Mutagenesis kit (Stratagene, Cedar Creek, TX, USA), according to the manufacturer’s instructions. miR-486-5p binding sites were point mutated as following: 606-614→ UACAGGA to UGCGGAA and 1993–2000 → GUACAGG to GCAUAUG. Sequences of LINC02802, LINC02802-MUT, MAML3 and MAML3-MUT were cloned into pmirGLO dual luciferase miRNA target expression vector (Promega, San Luis Obispo, CA, USA) and used in the luciferase assays.

### 4.9. Transient Transfections

To overexpress or inhibit miRNAs, HUVECs were transfected with mirVana™ miRNA mimic or mirVana™ miRNA inhibitor, or with the corresponding control (ThermoFisher Scientific, Waltham, MA, USA), at a final concentration of 90 nM by using RNAiMAX lipofectamine in Optimem medium (ThermoFisher Scientific, Waltham, MA, USA), according to the manufacturer’s protocol. Knockdown of MAML3( ) and LINC02802 was performed by transfecting ON-TARGET plus siRNAs into HUVECs by using RNAiMAX lipofectamine in Optimem medium (ThermoFisher Scientific, Waltham, MA, USA), according to the manufacturer’s protocol. Over-expression of MAML3 and LINC02802 was performed by transfecting pcDNA3.1 MAML3 and pcDNA3.1 LINC02802, respectively, into ECs by using lipofectamine 3000 in Optimem medium (ThermoFisher Scientific, Waltham, MA, USA), according to the manufacturer’s protocol. Transfection medium was replaced after overnight incubation. Effective over-expression or inhibition was verified by RT-qPCR, as described. Subsequent assays were performed 24 h post-transfection. siRNAs against LINC02802 were custom synthesized by Horizon Discovery Ltd. ((SiLINC02802-3: GCAACUUAGCCUAGCAUUAU; siLINC02802-2GUGAAAGGCUGCUGGCCAUU)). ON-TARGET plus Human MAML3 siRNA Set of 4 (LQ-013813-01-0010) was purchased from Horizon Discovery Ltd. ON-TARGETplus Control (ON-TARGETplus non-targeting siRNAs control#3, D-001810-10) was used as the control for every siRNA experiment.

### 4.10. Dual-Luciferase Reporter Assay

Next, 3x105 ECs were co-transfected with pMIR-LINC02802 WT or pMIR-LINC02802 MUT and pMIR-MAML3 WT or pMIR-LINC02802 MUT, together with miR-486-5p miRNA mimic or miRNA Mimic Negative Control #1 (90 nM), using RNAiMAX lipofectamine (Ther-moFisher Scientific, Waltham, MA, USA). Lysates were collected 24 h after transfection and Firefly and Renilla luciferase activities were measured by using the dual-luciferase reporter assay system (Promega, San Luis Obispo, CA, USA) in a Bio-Tek Synergy HT Multi-Detection Microplate Reader (MTX Labsystems, Vienna, VA, USA).

### 4.11. Western Blot Analysis

Total proteins were obtained by extraction with a cell lysis buffer containing 0.125 M Tris-HCl, pH 6.8, 4% SDS, 20% glycerol, and quantified by BCA assay (ThermoFisher Scientific, Waltham, MA, USA). Equal amounts of proteins per sample were separated by SDS-PAGE (ThermoFisher Scientific, Waltham, MA, USA) and subsequently blotted to a PVDF membrane (Biorad, Hercules, CA, USA). Membranes were incubated with specific primary antibodies and the corresponding HRP-conjugated secondary antibodies. Immunoreactive proteins were visualized by enhanced chemiluminescence (ECL) system and acquired using a ChemiDoc Touch Gel Imaging System (Biorad, Hercules, CA, USA). Images were analyzed with Image Lab software 5.2.1 (Biorad, Hercules, CA, USA). Antibodies used were anti-MAML3 (cat # PA5-13678, ThermoFisher Scientific, Waltham, MA, USA) and anti- beta I Tubulin [EPR16778] (cat # ab179511, Abcam).

### 4.12. Statistical Analysis

All statistical analyses were performed using the GraphPad Prism 6 software (GraphPad, San Diego, CA, USA). Pooled data are expressed as the mean ± SEM. n represents the number of independent biological replicates, performed with different HUVEC batches. Significance was determined by using unpaired Student’s *t*-test (two-tailed), assuming a normal distribution. *p* < 0.05 was considered significant. Specific details for each experiment can be found in the corresponding figure legend.

## 5. Conclusions

Our study reveals that LINC02802 functions as an miRNA sponge to positively regulate MAML3 expression through sponging miR-486-5p, and subsequently promotes sprouting angiogenesis in an ECs spheroids model.

## Figures and Tables

**Figure 1 ijms-23-01653-f001:**
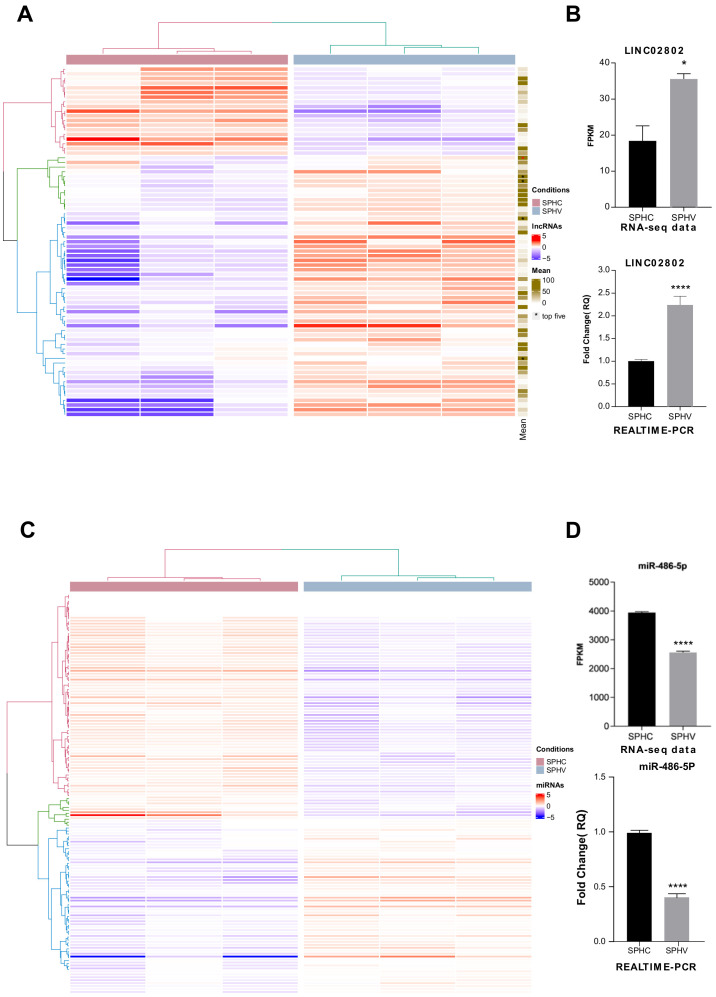
Identification of lncRNA and microRNA involved in the regulation of SA (**A**) Heatmap showing DE lincRNAs in SPHV versus SPHC with log2FC > 0.5 and FDR < 0.05. Color bar indicates log2FC. The mean column showed the expression rate within our experimental condition. (**B**) LINC02802 expression in SPHC and SPHV from RNA-sequencing data and LINC02802 expression in SPHC and SPHV measured by RT-qPCR. Data are represented as mean ± SEM of *n* = 3 experiments; *, *p* < 0.05. ****, *p* < 0.0001. (**C**) Heatmap showing DE microRNAs in SPHV versus SPHC with log2FC > 0.5 and FDR < 0.05. Color bar indicates log2FC. (**D**) mir-486-5p expression in SPHC and SPHV from RNA-sequencing data and mir-486-5p expression in SPHC and SPHV measured by RT-qPCR. Data are represented as mean ± SEM of *n* = 3 experiments; ****, *p* < 0.0001.

**Figure 2 ijms-23-01653-f002:**
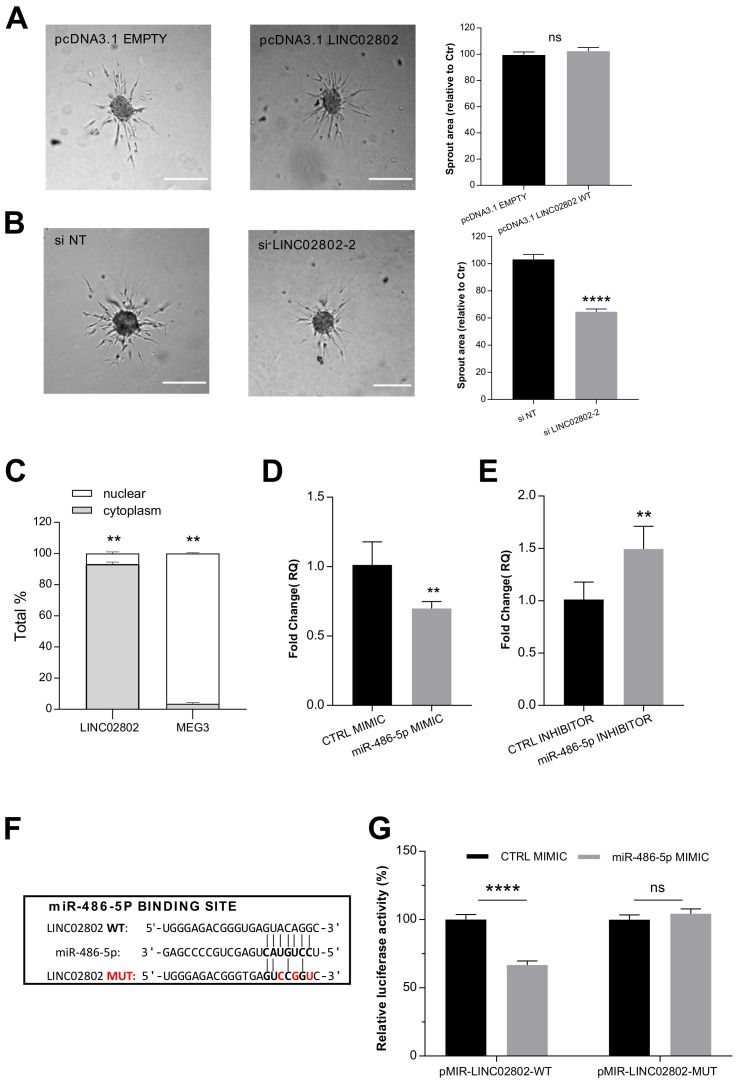
LINC02802 acts as a sponge for miR-486-5p (**A**) Sprouting assay performed under LINC02802 over-expression and relative quantification of sprout area. (**B**) Sprouting assay performed with ECs transfected with si LINC02802-2 and relative quantification of sprout area. Scale bars, 200 mm. Total sprout area relative to control spheroids is shown. Data are represented as mean ± SEM from *n* = 3 experiments; ****, *p* < 0.0001; ns, not significant. (**C**) Subcellular fractionation assay measured the localization of LINC02802 in ECs cells. Data are represented as mean ± SEM of *n* = 3 experiments; **, *p* < 0.01. (**D**) RT-qPCR analysis of relative LINC02802 levels in CTRL mimic or miR-486-5p mimic transfected ECs and (**E**) in CTRL inhibitor or miR-486-5p inhibitor. Data are represented as mean ± SEM from *n* = 3 experiments; **, *p* < 0.01. (**F**) Representation of the binding site for miR-486-5p on LINC02802 sequence and relative mutation. Nucleotides that were mutated to disrupt the interaction are indicated in red; (**G**) Luciferase reporter constructs containing wild type or mutated miR-486-5p binding sites on LINC02802 were co-transfected with miR-486-5p mimic or CTRL mimic into ECs. Luciferase signal was assayed at 24 h post-transfection and normalized to Renilla luciferase activity. Data are represented as mean ± SEM from *n* = 3 experiments; ****, *p* < 0.0001, ns, not significant.

**Figure 3 ijms-23-01653-f003:**
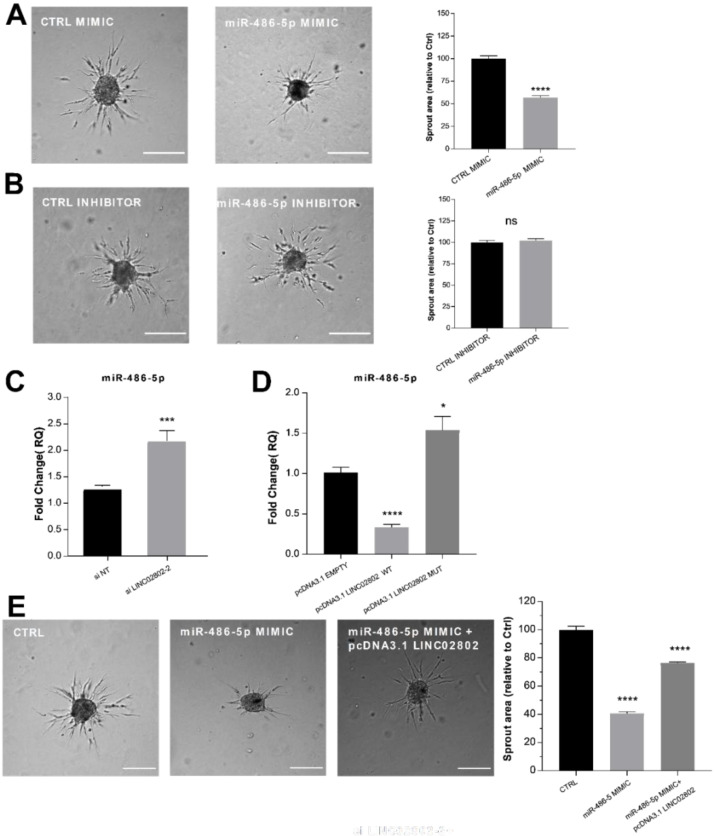
LINC02802 counteracting the anti-angiogenic effect of miR-486-5p (**A**) Sprouting assay performed with ECs transfected with miR-486-5p mimic and relative quantification of sprout area. (**B**) Sprouting assay performed with ECs transfected with miR-486-5p inhibitor and relative quantification of sprout area. Scale bars, 200 mm. Total sprout area relative to control spheroids is shown. Data are represented as mean ± SEM from *n* = 3 experiments; ****, *p* < 0.0001; ns, not significant. (**C**) RT-qPCR of relative miR-486-5p expression in ECs transfected with si LINC02802-2 and (**D**) in ECs transfected with pcDNA3.1 LINC02802 WT or pcDNA3.1 LINC02802 MUT, respectively. Data are represented as mean ± SEM from *n* = 3 experiments; *, *p* < 0.05 ***, *p* < 0.001; ****, *p* < 0.0001; (**E**) Sprouting assay performed with ECs transfected with miR-486-5p mimic alone or co-transfected with pcDNA3.1 LINC02802 WT and relative quantification of sprout area. Scale bars, 200 mm. Total sprout area relative to control spheroids is showed. Data are represented as mean ± SEM from *n* = 3 experiments; ****, *p* < 0.0001.

**Figure 4 ijms-23-01653-f004:**
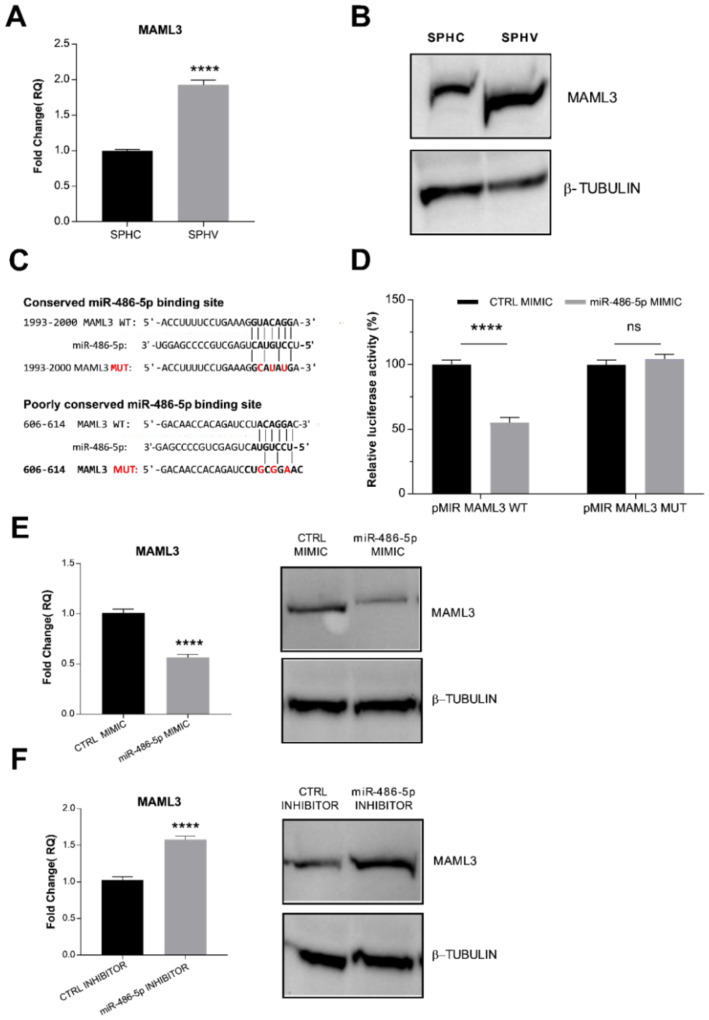
miR–486–5p directly targets MAML3 (**A**) RT-qPCR analysis of MAML3 in spheroids. Data are represented as mean ± SEM from *n* = 3 experiments; ****, *p* < 0.0001. (**B**) Detection of MAML 3 by Western blot analysis. Image are representative of three different experiments. (**C**) Representation of the two binding sites for miR-486-5p on MAML3 3′-UTR. Nucleotides that were mutated to disrupt the interaction are indicated in red. (**D**) Luciferase reporter assay in ECs co-transfected with miR-486-5p mimic together with pcDNA3.1 MAML3 WT or the mutant form. Luciferase signal was assayed at 24 h post-transfection and normalized to Renilla luciferase activity. Data are represented as mean ± SEM from *n* = 3 experiments; ****, *p* < 0.0001, ns, not significant. (**E**) RT-qPCR and Western blot analysis of MAML3 expression in ECs transfected with miR-486-5p mimic. (**F**) RT-qPCR and Western blot analysis of MAML3 expression in ECs transfected with miR-486-5p inhibitor. Data are represented as mean ± SEM from *n* = 3 experiments; ****, *p* < 0.0001. Western images are representative of three different experiments.

**Figure 5 ijms-23-01653-f005:**
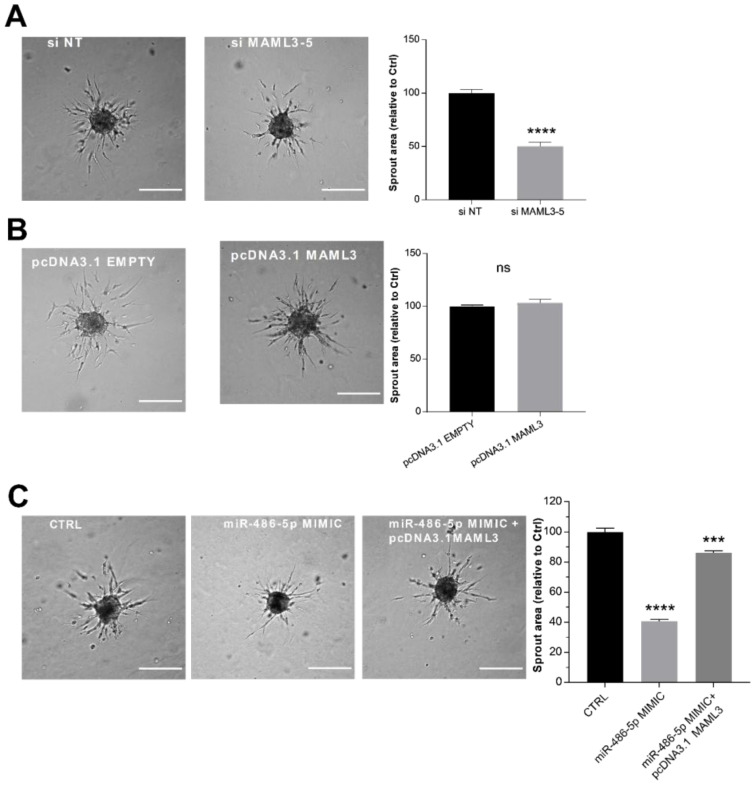
MAML3 over-expression counteracts the inhibitory effect of miR-486-5p on SA (**A**) Sprouting assay performed with ECs transfected with an siRNA against MAML3, and its quantification. (**B**) Sprouting assay performed with ECs transfected with pcDNA3.1 MAML3, and corresponding quantification. (**C**) Sprouting assay performed with ECs transfected with miR-486-5p mimic alone or co-transfected with miR-486-5p mimic and pcDNA3.1 MAML3. Scale bars = 200 µm. Total sprout area relative to control spheroids is showed. Data are representative of three independent experiments and shown as mean ± SE; ****, *p* < 0.0001; ***, *p* < 0.001.

**Figure 6 ijms-23-01653-f006:**
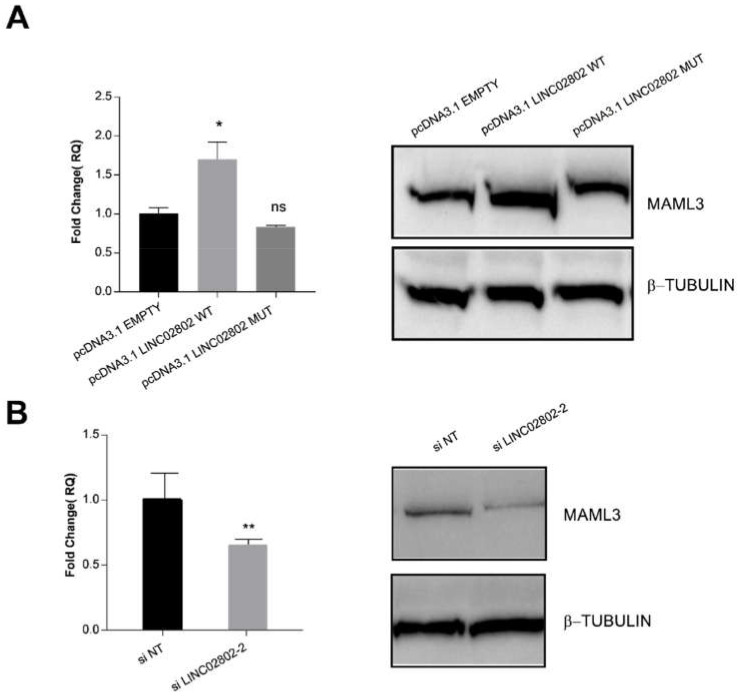
LINC02802 indirectly modulates MAML3 expression (**A**) RT-qPCR and Western blot analysis of relative MAML3 expression in ECs transfected with LINC02802 wild type or the mutated one. (**B**) RT-qPCR and Western blot analysis of relative MAML3 expression in ECs transfected with si_LINC02802. Data are represented as mean ± SEM from *n* = 3 experiments; **, *p* < 0.01; *, *p* < 0.05; ns, not significant. Blots are representative of three experiments performed with similar results.

**Figure 7 ijms-23-01653-f007:**
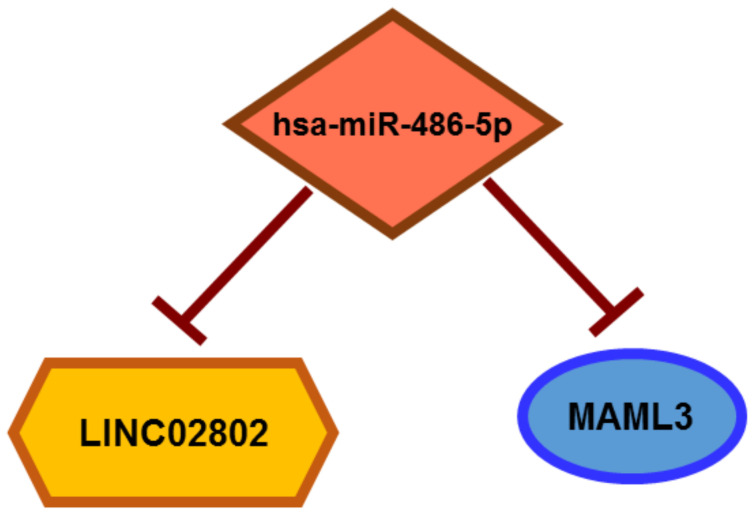
Schematic representation of LINC02802, mir-486-5p and MAML3 circuit.

## Data Availability

Data used in this study were originally published in [10] and are available as GSE115817 and GSE115954 in the GEO database.

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
