# Peer review of "Long Non-Coding RNA LINC02802 Regulates In Vitro Sprouting Angiogenesis by Sponging microRNA-486-5p"

_ijms, 2022, doi:10.3390/ijms23031653_

Round 1
Reviewer 1 Report
In this study the authors assessed the transcriptomic profile of VEGF-A-activated endothelial cells and identified LINC02802 as key-regulator of endothelial sprouting regulation. LINC02802 acted as a ceRNA in the post- transcriptional regulation of Mastermind-like-3 (MAML3) gene expression through competitive binding with miR-486-5p. The experiments are solid and are well conducted. I have a very few comments
- The Introduction of the article is way to long. Please shorten it a bit.
- Please adjust the following sentence. It is very unclear. 'To this task, we explored by a bioinformatic approach the miRNAs possibly interacting with LINC02802 by considering a combination of sequence-based predicting algorithms (see Methods) and considering only the miRNAs differentially expressed between SPHV and SPHC (Figure 1C).'
- I could not find the figure you refer to within the following sentence: 'The efficiency of overexpression and knockdown of miR-486-5p were evaluated by RT-qPCR (Figure supplementary 1C-1D).' Please elaborate a bit more on the qPCR efficiency difference since it refers to the probability that miR-486 regulates the LINC02802.
- Figure 5B: how come MAML3 overexpression did not affect the sprouting activity? Please explain.
Reviewer 2 Report
In the submitter paper, Rosano et al. demonstrates that LINC02802 act as a sponge for mir-486-5p and as a ceRNA for MAML3 to regulate sprouting angiogenesis in vitro. The paper addresses an important issue of the network of dependence between miRNAs, miRNA targets and lncRNA acting as sponges that contribute to the observed sprouting. The presented results are novel, as the interactions between miR-486, LINC02802 and MAML3 were not shown before, and function of LINC02802 was not known. The paper is clearly written, and well-organized. I only have minor issues:
- The Figures 1 and 2, descriptions in the graphs are illegible. Please provide graphs with higher resolution.
- In a legend of Figure 2 and in line226, it is incorrectly written that LINC02802 is a ceRNA for miR-486. LINC02802 can be a ceRNA for MAML3, as the authors showed that they are directly regulated by the same miRNA. LINC02802 could only be a sponge for miR-486. The term ceRNA is, however, properly used in other sections in the manuscript.
- lncRNA that act as sponges for miRNAs often have multiple binding sites, so that they bind more miRNA molecules that protein coding targets. Does LINC02802 contain more that one miR-486 binding site?
- Would LINC02802 be a ceRNA for MAML3 and sponge for miR-486 in physiological conditions ( and levels)?
- It would beneficial for understanding if the authors include simple scheme showing the interactions between miR-486, LINC02802 and MAML3.
Methods section:
- Please include sequences of siRNA against LINC02802 and MAML3 and control, sequences of custom Taqman Assays for LINC02802, and sequences used for overexpression of LINC02802 and MAML3.
- To my knowledge Oligotex is a method to isolate polyadenylated RNA. Please include information about the protocol for nuclear/cytoplasmic fraction, and control RNA for cytoplasmic fraction.
